# Concussion-Associated Polygenic Profiles of Elite Male Rugby Athletes

**DOI:** 10.3390/genes13050820

**Published:** 2022-05-04

**Authors:** Mark R. Antrobus, Jon Brazier, Peter C. Callus, Adam J. Herbert, Georgina K. Stebbings, Praval Khanal, Stephen H. Day, Liam P. Kilduff, Mark A. Bennett, Robert M. Erskine, Stuart M. Raleigh, Malcolm Collins, Yannis P. Pitsiladis, Shane M. Heffernan, Alun G. Williams

**Affiliations:** 1Department of Sport and Exercise Sciences, Manchester Metropolitan University Institute of Sport, Manchester Metropolitan University, Manchester M1 7EL, UK; j.brazier2@herts.ac.uk (J.B.); peter.callus@stu.mmu.ac.uk (P.C.C.); g.stebbings@mmu.ac.uk (G.K.S.); praval.khanal@gmail.com (P.K.); a.g.williams@mmu.ac.uk (A.G.W.); 2Sport and Exercise Science, University of Northampton, Northampton NN1 5PH, UK; 3Department of Psychology and Sports Sciences, University of Hertfordshire, Hatfield AL10 9AB, UK; 4Research Centre for Life and Sport Sciences (C-LaSS), School of Health Sciences, Birmingham City University, Birmingham B15 3TN, UK; adam.herbert@bcu.ac.uk; 5School of Medicine and Clinical Practice, University of Wolverhampton, Wolverhampton WV1 1LY, UK; stephen.day@wlv.ac.uk; 6Applied Sports Science Technology and Medicine Research Centre (A-STEM), College of Engineering, Swansea University, Swansea SA1 8EN, UK; l.kilduff@swansea.ac.uk (L.P.K.); markbennettanalytics@gmail.com (M.A.B.); s.m.heffernan@swansea.ac.uk (S.M.H.); 7Research Institute for Sport & Exercise Sciences, Liverpool John Moores University, Liverpool L3 3AF, UK; r.m.erskine@ljmu.ac.uk; 8Institute of Sport, Exercise and Health, University College London, London WC1E 6BT, UK; 9Cardiovascular and Lifestyle Medicine Research Group, CSELS, Coventry University, Coventry CV1 5FB, UK; ac8510@coventry.ac.uk; 10Health through Physical Activity, Lifestyle and Sport Research Centre (HPALS), Department of Human Biology, and the International Federation of Sports Medicine (FIMS) Collaborative Centre of Sports Medicine, University of Cape Town, Rondebosch, Cape Town 7701, South Africa; malcolm.collins@uct.ac.za; 11FIMS Reference Collaborating Centre of Sports Medicine for Anti-Doping Research, University of Brighton, Brighton BN20 7SP, UK; y.pitsiladis@brighton.ac.uk; 12Centre for Exercise Sciences and Sports Medicine, FIMS Collaborating Centre of Sports Medicine, Piazza L. de Bosis 6, 00135 Rome, Italy

**Keywords:** rugby, genotype, concussion, brain, polymorphism, genetics

## Abstract

Due to the high-velocity collision-based nature of elite rugby league and union, the risk of sustaining a concussion is high. Occurrence of and outcomes following a concussion are probably affected by the interaction of multiple genes in a polygenic manner. This study investigated whether suspected concussion-associated polygenic profiles of elite rugby athletes differed from non-athletes and between rugby union forwards and backs. We hypothesised that a total genotype score (TGS) using eight concussion-associated polymorphisms would be higher in elite rugby athletes than non-athletes, indicating selection for protection against incurring or suffering prolonged effects of, concussion in the relatively high-risk environment of competitive rugby. In addition, multifactor dimensionality reduction was used to identify genetic interactions. Contrary to our hypothesis, TGS did not differ between elite rugby athletes and non-athletes (*p* ≥ 0.065), nor between rugby union forwards and backs (*p* = 0.668). Accordingly, the TGS could not discriminate between elite rugby athletes and non-athletes (AUC ~0.5), suggesting that, for the eight polymorphisms investigated, elite rugby athletes do not have a more ‘preferable’ concussion-associated polygenic profile than non-athletes. However, the *COMT* (rs4680) and *MAPT* (rs10445337) GC allele combination was more common in rugby athletes (31.7%; *p* < 0.001) and rugby union athletes (31.8%; *p* < 0.001) than non-athletes (24.5%). Our results thus suggest a genetic interaction between *COMT* (rs4680) and *MAPT* (rs10445337) assists rugby athletes in achieving elite status. These findings need exploration vis-à-vis sport-related concussion injury data and could have implications for the management of inter-individual differences in concussion risk.

## 1. Introduction

Sport-related concussion has been defined as a traumatic brain injury (TBI) induced by external forces [1]. It has been reported that over a playing career, ~80% of rugby (league and union) players will experience at least one concussion [2]. In male elite rugby union (RU), concussion has been the most common injury in the English Premiership since 2011 (accounting for 21% of all injuries from the 2014 to 2019 seasons) [3]. In elite male rugby league (RL), concussion accounted for 29% of all injuries in illegal play and 9% of all injuries in legal play [4]. Sustaining a prior concussion increases the risk of subsequent time-loss injuries and repeated concussions [5,6,7,8]. There is a growing concern about the potential short and long-term neurodegenerative consequences associated with concussion, such as chronic post-concussion syndrome, cognitive impairment, forms of dementia, migraines, sleep dysfunction and anxiety [1,9,10,11,12,13]. Several genetic factors have previously been associated with inter-individual variability in traumatic brain injury incidence and severity [14].

Molecular pathophysiological investigations show that a concussive event can trigger a neurometabolic cascade resulting in altered gene expression and neuronal dysfunction within the brain [15]. Inter-individual variability means that the severity of concussion in rugby can range from 2 days to >84 days absence (period from injury to availability for match selection) but typically ranges from ranges 9–21 days [8,10,16,17,18]. To better understand and manage the inter-individual variability in injury occurrence and outcomes following concussion, the main risk factors must be identified. One such factor is genetic predisposition, as the interaction of multiple genes in a polygenic manner could reflect the complex pathophysiology of incidence and recovery from concussion [19]. Heritability of concussion has not been determined, but it is likely that a substantial genetic component exists for concussion risk and recovery, as heritability of brain structure is shown to be ~90% and cognitive performance ~60% [20,21,22,23]. Previous candidate gene studies have identified potential genetic risk factors associated with the risk of concussion and recovery [6,7,24,25,26,27,28,29,30,31,32,33]. Those genetic variants influencing concussion risk and recovery may confer an advantage/disadvantage for rugby athletes by affecting the ability to train and compete and thus advance their careers, and investigating this could provide additional information to support the management of the cumulative effects of concussions [14]. 

The polymorphisms that have been examined in this study have some of their basic functions summarised in Table 1. Regarding concussion in particular, the ε4 allele of a*polipoprotein* (*APOE*) gene could be responsible for up to 64% of the ‘hazardous influence’ of TBI [24] and athletes who possess the ε4 allele suffered prolonged physical and cognitive symptomatic responses to concussion [25]. Carriers of the *APOE* promoter T allele have a three to eight-fold greater risk of experiencing repeated concussions [5,7], and TT genotype carriers were observed to experience unfavourable outcomes post-TBI [26]. From the *microtubule-associated protein tau* (*MAPT*) gene, the TT genotype has been weakly associated with a greater risk of repeated concussion [6,7]. The *Nitric oxide synthase* (*NOS3)* gene C allele has been associated with lower cerebral blood flow in patients with severe TBI [27]. The T allele of the *ankyrin repeat and kinase domain-containing 1* (*ANKK1*) gene has been associated with worse measures of learning, working memory and response latencies post-TBI [28,29,30]. In addition, the *brain-derived neurotrophic factor (BDNF)* gene Met/Met homozygotes have been reported to be at a higher risk of sustaining a concussion than Val/Val homozygotes [31]. The *catechol-O-methyltransferase* (*COMT*) gene rs4680 Val allele carriers performed poorer on tests of executive function post-TBI [32], and Met carrying RU players have been reported to be approximately three-fold more likely to have a history of concussion [33]. Indeed, elite rugby athletes have ~1.4 times the odds of being Val/Val (GG) genotype compared to non-athletes [34], while other single nucleotide polymorphism (SNP) variants related to different injury types are also more frequent in elite rugby athletes [35].

The Total Genotype Score (TGS) has been used to indicate the extent of an individual’s genetic predisposition for athletic performance, muscle damage and disease risk [36,37,38,39,40,41,42]. Based on a genetic algorithm proposed by Williams and Folland [36], TGS can range from 0 to 100 and represents the number of ‘preferable’ genotypes an individual possesses for the phenotype in question. Previous hypothetical and experimental TGS studies indicate that athletes have higher TGS scores, thus possessing more ‘preferable’ polygenic profiles than non-athletes for performance, injury and potential disease [36,37,38,39,40,41,42,43,44,45]. The potential applications of this approach to concussion are attractive but have not yet been explored.

Therefore, the primary objective of this study was to investigate if concussion-associated polygenic profiles differ between elite rugby athletes and non-athletes. Based on prior literature, it was hypothesised that the elite rugby athletes would have a higher TGS than non-athletes, indicating a more ‘preferable’ polygenic profile with respect to concussion, and/or display gene–gene interactions that differ from non-athletes.

## 2. Materials and Methods

### 2.1. Participants

As part of the ongoing RugbyGene project [46], a total of 1357 individuals were recruited and provided written informed consent to participate in the present study. An a priori calculation for 80% power to detect a small effect size (w) of 0.1 indicated > 785 participants were required. The total sample comprised elite 635 Caucasian male rugby athletes (mean (standard deviation) height 1.85 (0.07) m, mass 102 (12) kg, age 29 (7) years), including 66.4% British, 11.4% Irish, 9.5% Italian, 8.9% South African, and 3.8% of other nationalities, and 722 Caucasian non-athletes (48% male, mean (standard deviation) height 1.70 (0.10) m, mass 73 (13) kg, age 41 (23) years), including 97.6% British and 2.4% other nationalities. Athletes were considered elite if they had competed regularly (>5 matches) since 1995 in the highest professional league in the UK, Ireland or South Africa for RU or the highest professional league in the UK for RL [47]. In total, 53.5% of the RU athletes had competed at the international level for a “high performance union” (Regulation 16, https://www.world.rugby/organisation/governance/regulations/reg-16 (accessed on 1 February 2022)) and 45.8% of RL athletes had competed at the international level. As the majority of athletes (534) competed in RU, they were also divided into forwards (304) and backs (230) for comparison (to detect an effect size (w) of 0.13 required >464 participants). Ethical approval was granted by the ethics committees of Manchester Metropolitan University, Glasgow University, University of Cape Town and University of Northampton, and all experimental procedures complied with the Declaration of Helsinki [48].

### 2.2. Procedures

#### 2.2.1. Sample Collection

Procedures were consistent with those described previously [35,47,49]. Blood (70.4% of all samples), buccal swabs (15.4%) or saliva (14.2%) samples were obtained. Blood was drawn from a superficial forearm vein into EDTA tubes, saliva samples were collected into Oragene DNA OG-500 tubes (DNA Genotek, Ottawa, ON, Canada) and sterile buccal swabs (Whatman OmniSwab, Springfield Mill, UK) were rubbed against the buccal mucosa of the cheek for ∼30 s.

*DNA isolation and genotyping.* DNA isolation and genotyping were performed in the Manchester, Glasgow and Cape Town laboratories. The majority of samples were processed and genotyped in the Manchester laboratory. There are some differences between protocols, summarised below.

In Manchester and Glasgow, DNA isolation was performed with the QIAamp DNA Blood Mini kit and spin column protocol (Qiagen, West Sussex, UK). Briefly, 200 μL of whole blood was lysed and incubated, the DNA washed, and the eluate stored at 4 °C. In Cape Town, using a different protocol [50], samples were lysed and centrifuged, the DNA washed, and samples stored at −20 °C. DNA isolated in Cape Town was genotyped in Glasgow.

Genotyping for eight polymorphisms (see Genotyping assays) was performed using two protocols. Protocol one: Approximately 40% of samples were genotyped using a StepOnePlus (Applied Biosystems, Paisley, UK) as previously described [47] with variations to thermocycling conditions depending on reagents used. Protocol two: Approximately 60% of samples were genotyped by combining 2 μL of GTXpress Master Mix (2×) (Applied Biosystems), 0.2 μL of 20× Fast GT Sample Loading Reagent (Fluidigm, Cambridge, UK), 0.2 μL of H_2_O and 1.6 μL of purified DNA. Furthermore, 1.78 μL of assay (20×) (Applied Biosystems), 1.78 μL of 2× Assay Loading Reagent (Fluidigm) and 0.18 μL of ROX reference dye (Invitrogen, Paisley, UK) were combined. An integrated fluid circuit controller RX (Fluidigm) mixed samples and assays using a Load Mix (166×) script. PCR was performed using a real-time FC1 Cycler (Fluidigm) GT 192 × 24 Fast v1 protocol. The 192 × 24 microchip plate was placed into the EP1 Reader (Fluidigm) for end-point analysis using Fluidigm SNP genotyping analysis software. Duplicates of all samples were in 100% agreement for both protocols.

#### 2.2.2. Genotyping Assays

For *ANKK1* (rs1800497), *APOE* (rs429358, rs7412 and rs405509), *BDNF-AS* (rs6265), *COMT* (rs4680), *MAPT* (rs10445337) and *NOS3* (rs2070744), the appropriate TaqMan assays were utilised (Applied Biosystems). *APOE* ε2/ε3/ε4 data were derived from rs429358 and rs7412 [51]. The TaqMan assay context sequence for each polymorphism, with VIC/FAM highlighted in **bold** and concussion-associated risk alleles underlined (although for some the prior evidence of risk is controversial), were: *ANKK1* (rs1800497) TGGTC[**A/G**]AGGCA, *APOE* (rs429358) ACGTG[**C/T**]GCGGC*, APOE* (rs7412) AGAAG[**C/T**]GCCTG*, APOE* (rs405509) GTCTG[**G/T**]ATTAC*, BDNF-AS* (rs6265) TATCA[**C/T**]GTGTT*, COMT* (rs4680) CTGGC[**A/G**]TGAAG*, MAPT* (rs10445337) TCACT[**C/T**]CCCGA*, NOS3* (rs2070744) CTGGC[**C/T**]GGCTGA.

### 2.3. Calculation of TGS

To quantify the combined influence of the candidate polymorphisms (Table 2), an additive TGS algorithm was utilised [36] based on the assumption of codominant allele effects. For bi-allelic polymorphisms, the homozygote genotypes with the lower concussion risk and ‘preferable’ outcome according to prior literature were allocated a genotype score of 2, heterozygote genotypes scored 1 and the ‘non-preferable’ homozygote genotypes scored 0. *APOE* is a tri-allelic (ε2, ε3, ε4) polymorphism—two C/T SNPs at residues 112 (rs429358) and 158 (rs7412) produce six possible genotypes (ε2/ε2, ε2/ε3, ε2/ε4, ε3/ε3, ε3/ε4, ε4/ε4). A score of 0 was allocated for ε4 allele possession (ε4+) and a score of 2 was allocated for non-possession of a ε4 allele (ε4−) (no score of 1 allocated). A TGS of 100 represents the ‘perfect’ polygenic profile for low concussion risk and favourable outcome, for the eight SNPs (seven genotype scores) examined, while 0 represents the ‘worst’ possible profile for concussion risk and outcome.
TGS = (100/14) ∗ *ANKK1*_rs1800497_ + *APOE*_rs429358, rs7412_ + *APOE*_rs405509_ + *BDNF-AS*_rs6265_, *COMT*_rs4680_ + *MAPT*_rs10445337_ + *NOS3*_rs2070744_

In addition, a TGS algorithm determined only by observed genotype frequencies in elite rugby athletes was also calculated, wherein three of the seven genotype scores allocated had different values that reflected the prior evidence [34] (*APOE* rs405509 TT = 2, GT = 1, GG = 0, *COMT* rs4680 GG = 2, GA = 1, AA = 0 and *NOS3* rs2070744 TT = 2, TC = 1, CC = 0).

### 2.4. Data Analysis 

SPSS for Windows version 26 (SPSS, Chicago, IL, USA) software was used for analysis. TGSs of athletes and non-athletes were compared using independent t-tests, as were height and body mass. Pearson’s χ^2^ tests were utilised to compare genotype frequencies in upper and lower TGS quartiles of all rugby athletes vs. non-athletes, RU athletes vs. non-athletes, RL athletes vs. non-athletes, RU Forwards vs. non-athletes, RU Backs vs. non-athletes and RU Forwards vs. RU Backs. Additionally, receiver operating characteristic (ROC) curves and area under the curve (AUC) were used to evaluate the ability of the TGS to correctly distinguish between athletes (including positional groups) and non-athletes [52]. Multifactor dimensionality reduction (https://sourceforge.net/projects/mdr/, (accessed on 1 February 2022)) software was used to identify SNP–SNP epistasis interactions [53] that distinguish between athlete groups and non-athletes. α was set at 0.05.

## 3. Results

### 3.1. TGS

Genotype frequencies were in the Hardy–Weinberg equilibrium for all polymorphisms in the non-athlete and athlete groups. Athletes (all male) were taller and heavier (*p* < 0.001) than male non-athletes. No participant had a TGS of zero or 100 (range was 21.4–92.9). Only one rugby athlete and one non-athlete possessed the highest observed TGS of 92.9. Similarly, five rugby athletes (0.8%) and just one non-athlete possessed the lowest observed TGS of 21.4. Collectively, 76.7% of rugby athletes and 77.2% of non-athletes had a TGS > 50.

There was no difference in TGS between any rugby athlete group (all rugby athletes, RU, RL, RU forwards, RU backs) and the non-athlete group (*p* ≥ 0.769). Mean (SD) and kurtosis statistics of TGSs are reported in Table 2, and frequency distributions of rugby athletes and non-athletes are shown in Figure 1A. Similarly, there was no difference in TGS between RU forwards and backs (*p* = 0.842).

When the numbers of athletes (including discrete groups) and non-athletes in the upper and lower 25% of TGS were compared, no significant differences were found (Table 3). Similarly, there were no differences between RU forwards and backs in terms of their presence in the upper and lower quartiles of TGS (*p* = 0.668). ROC AUC analysis confirmed the TGS algorithm could not identify elite rugby athlete status (AUC = 0.504; 95% CI = 0.470–0.538; *p* = 0.800; Figure 1B). There was also no ability to distinguish between athletes and non-athletes when discrete groups of athletes were considered (RU vs. non-athletes, RL vs. non-athletes, RU Forwards vs. non-athletes, RU Backs vs. non-athletes; Table 3).

Even when using a TGS algorithm using genotype scores determined solely by genotype frequencies we observed in elite rugby athletes, there was no difference between all rugby athletes and non-athletes (*p* = 0.065; Figure 2), nor was there a difference between the numbers of athletes and non-athletes in the upper and lower TGS quartiles (*p* = 0.144). ROC AUC analysis again demonstrated that the data could not correctly distinguish elite rugby athletes from non-athletes (AUC = 0.532; 95% CI = 0.497–0.567, *p* = 0.067).

### 3.2. SNP Epistasis

Multifactor dimensionality reduction analysis identified an SNP–SNP interaction of *COMT* rs4680 and *MAPT* rs10445337 polymorphisms that best predicted elite athlete status (testing accuracy 0.531; cross-validation consistency 9/10). There was a greater frequency of the *COMT-MAPT* G-C allele combination in all rugby athletes (31.7%; OR = 1.43, 95% CI = 1.12–1.81) and RU athletes (31.8%; OR = 1.44, 95% CI = 1.12–1.84) than non-athletes (24.5%; both comparisons *p* < 0.001) (Figure 3).

## 4. Discussion

In this present study, we investigated suspected concussion-associated polygenic profiles for determining elite status in rugby. This was the first use of elite rugby athlete data, TGS models and SNP–SNP epistasis interactions to determine whether a concussion-associated polygenic profile is more suitable for achieving elite status in the high concussion risk environment of rugby. For the eight suspected concussion-associated genetic variants used in the TGS algorithm, there was no difference in elite rugby athlete and positional subgroup TGSs compared to non-athletes. However, multifactor dimensionality reduction analysis found a 2-SNP model of *COMT* (rs4680) and *MAPT* (rs10445337) G-C allele combination produced the best model for predicting elite athlete status (testing accuracy 0.531; cross-validation consistency 9/10).

Our data show that mean concussion-associated TGS is approximately 56–57 for both rugby athletes and non-athletes, based on the eight SNPs we studied. This finding indicates that, for these eight SNPs previously associated with concussion incidence and/or severity, elite rugby athletes do not tend to have a more ‘preferable’ polygenic concussion-associated profile than non-athletes, thus not supporting our hypothesis. This contrasts with previous associations of physical performance-associated (as opposed to injury-associated) TGSs with elite athlete status of track and field athletes, rowers, cyclists and soccer athletes [38,40,41,44,54]. However, a musculoskeletal soft-tissue injury-associated TGS has recently been shown to differ between injured and non-injured soccer players [55].

It was anticipated that the TGS distributions of athletes and non-athletes might differ at their extremes. However, there was no difference between the upper and lower TGS quartiles in terms of the proportion of rugby athletes and non-athletes. The same large interindividual variability (TGS range = 21–93) was observed for both elite rugby athletes and non-athletes. This wide distribution of scores highlights the variable genetic potential with respect to concussion in the general population. Nevertheless, ~23% of elite rugby players possessed a concussion-associated TGS of 50 or less, which could indicate those athletes are more at risk of concussion and/or poorer outcome post-concussion due to possession of a ‘less preferable’ polygenic concussion profile.

We separately found no differences in genotype frequencies between elite rugby athletes and non-athletes for seven out of eight concussion-associated genetic variants [34]. That might have been because individual variants cannot represent the complexity of concussion risk and do not reflect SNP–SNP interactions, known as non-linear interaction or epistasis [53]. Here, we identify that the 2-SNP model of the *COMT* (rs4680) and *MAPT* (rs10445337) polymorphisms produced the best model to predict elite athlete status. The GC allele combination was more common in rugby athletes (~32%) than non-athletes (~25%). Previously, GG (Val/Val) carriers of *COMT* (rs4680) have been observed to have ~33% increased COMT activity when compared with AA (Met/Met) carriers, thus reducing dopamine levels in the prefrontal cortex region of the brain [56]. Lipsky et al. [23] observed that GG carriers had 40% poorer executive function than AA carriers post-TBI. Recently, it has been observed that elite rugby athletes have 1.4 times the odds of possessing the GG genotype of *COMT* (rs4680) compared to non-athletes [34]. In addition, Mc Fie et al. [33] observed that A allele carriers in a cohort of youth and professional South African RU players were approximately three-fold more likely to have a history of concussion. Considering the pleiotropic nature of *COMT* (rs4680), G carriers could possess greater stress resilience and reduced anxiety in competitive environments and be at lower risk of experiencing concussions but also be at risk of poorer cognitive function post-concussion [34,57,58]. The *MAPT* TT genotype (rs10445337) has been weakly associated with a greater risk of repeated concussion [6,7]. Mutations in *MAPT* have been shown to accelerate the aggregation of markers of neurotoxic hyperphosphorylated tau in response to repetitive concussions by 20–60% in animal studies and are associated with neurodegenerative diseases in humans [59,60]. Elite rugby athletes who possess the C allele could have a reduced risk of repeated concussion and potential neurodegenerative diseases. The GC allele combination could reduce the risk of experiencing concussions and provide a small advantage for attaining elite competitive status in the high concussion risk environment of competitive rugby. It should be noted that the SNP–SNP interaction analysis relies on data mining to identify the best genetic model to fit the data, potentially leading to overfitting. Cross-validation was utilised to compensate, although the 2-SNP model we identified should be investigated in other cohorts to confirm it.

The discriminatory power of the TGS is dependent upon the polymorphisms included, and the mathematical model utilised [61]. In both athletes and non-athletes, Ben-Zaken et al. [44] observed a higher mean TGS in a 2-SNP model than a 5-SNP model. However, the 5-SNP model provided greater discriminatory accuracy between groups [44]. In contrast, including many SNPs in a TGS model could reduce the explained variance. Thomaes et al. [62] reported that a 54-SNP model probably increased ‘background noise’ as all alleles were weighted equally, whereas—in reality—some variants will have larger effects on a phenotype than others. Adjustments to the weightings applied to each genetic variant in the algorithm could compensate, but a more extensive body of literature is required to apply relative weightings to different SNPs with confidence. The polymorphisms we included in the TGS are all reported to be associated with concussion (incidence, severity or recovery) or its related biological mechanisms [14].

In addition, our TGS gave all the SNPs equal weighting as we assumed allelic effects to be codominant and each SNP to have an equal additional effect, which may not uniformly be the case with respect to the pathophysiology of concussion in elite rugby athletes. The addition of polymorphisms in the TGS that do not influence the phenotype in question can decrease the discriminatory accuracy of the model [63]. In our study, only *COMT* (rs4680) has individually been previously associated with elite rugby athlete status [34]. Indeed, using a data-led TGS algorithm determined from previously observed genotype frequencies in elite rugby athletes [34], the eight suspected concussion-associated genetic variants we examined here were still unable to collectively distinguish athletes from non-athletes. Therefore, we cannot exclude the possibility that our TGS included polymorphisms potentially do not influence concussion risk in elite rugby athletes. Future studies will no doubt identify new candidate polymorphisms, and replication studies could indicate stronger associations between existing polymorphisms and concussion, which could be used to increase the accuracy of the algorithm. For example, a recent genome-wide-association study has identified two novel SNPs (*SPATA5* rs144663795 and *PLXNA4* rs117985931) associated with concussion [64]. There is also the question of whether a low number of concussions that have long recovery times or more numerous concussions that apparently resolve more quickly is more clinically problematic. Further GWASs and further replication studies of candidate gene approaches are needed to establish a TGS that quantifies estimated concussion risk effectively.

## 5. Conclusions

Concussion is a complex phenotype influenced by environmental factors and an individual’s genetic predisposition, and in the high concussion risk environment of elite rugby, genetically mediated resistance to aspects of concussion could be advantageous for career success and longevity. However, in contrast to our original hypothesis, a concussion-focused polygenic model could not discriminate between elite rugby athletes and non-athletes, although the large range of TGS scores could underpin the inter-individual variability in injury occurrence and outcomes following concussion. Nevertheless, epistasis analysis identified a genetic interaction of *COMT* (rs4680) and *MAPT* (rs10445337) G-C alleles as more common in elite rugby athletes, and carriage of these variants may affect stress resilience, behavioural traits and altered risk of concussion incidence and severity. It is possible that combining genetic data from multiple concussion-associated gene variants such as these could inform risk assessment and recovery from concussion in the future. Future studies should include polymorphisms for which strong associations with concussion have been newly observed to increase the accuracy of the model and potentially build towards a practical tool for concussion screening and management strategies in high concussion risk sports such as rugby.

## Figures and Tables

**Figure 1 genes-13-00820-f001:**
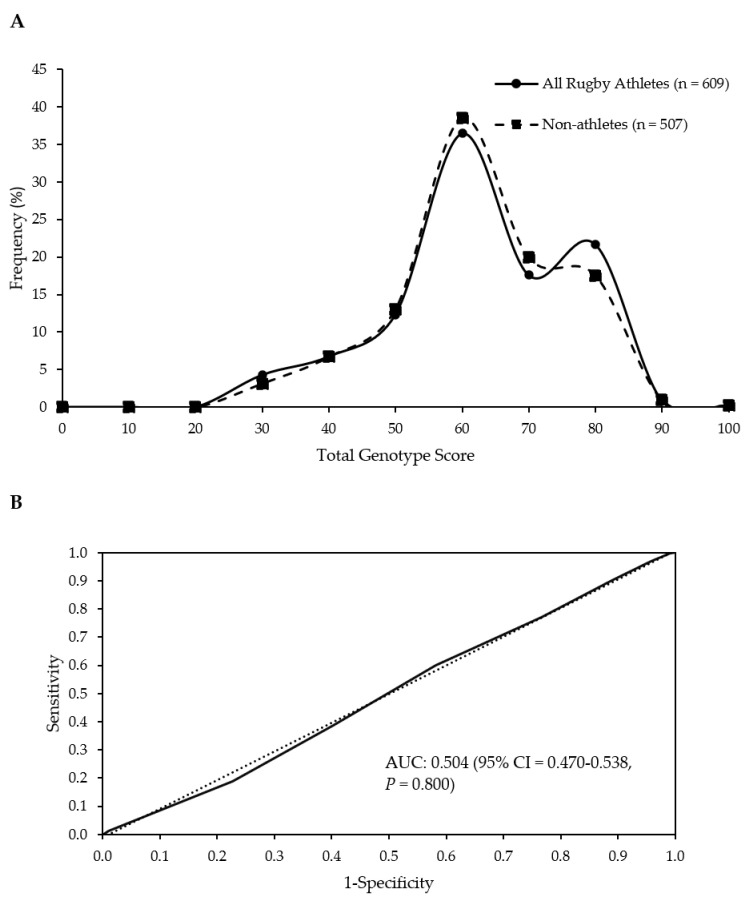
No difference in frequency distributions of the TGS of all athletes and non-athletes (*p* = 0.797 for comparison of means) (**A**). Receiver operating characteristic curve displays the inability of the TGS to discriminate elite rugby athletes from non-athletes. Dotted line = no discrimination. AUC; area under the curve (**B**).

**Figure 2 genes-13-00820-f002:**
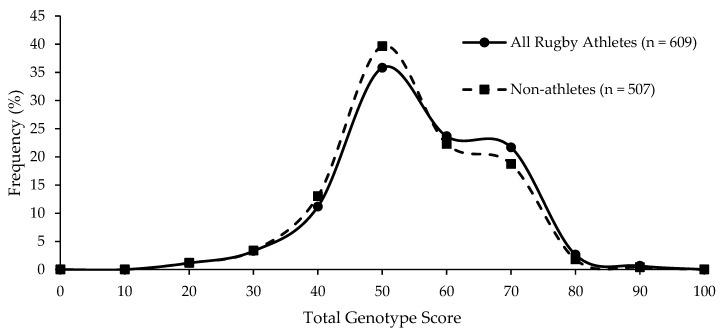
Similar frequency distribution of the data-led TGS for all athletes and non-athletes; *p* = 0.065 for difference in mean (SD) between all athletes (59.6 (12.4)) and non-athletes (58.4 (12.1)).

**Figure 3 genes-13-00820-f003:**
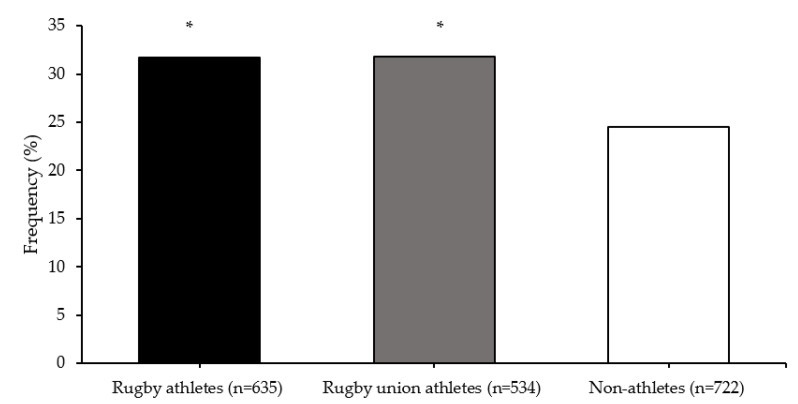
*COMT* (rs4680) and *MAPT* (rs10445337) G-C allele combination frequencies. * different from non-athletes (*p* < 0.001).

**Table 1 genes-13-00820-t001:** Summary of polymorphisms examined in this study.

Gene Name	Gene Abbreviation and Polymorphism Identifier	Alleles	Relevant Effects Associated with TBI
*Ankyrin repeat and kinase domain containing 1*	*ANKK1* rs1800497	A/G	The A allele has been associated with altered cognitive behavioural capacity via modulation of expression of D2 receptors.
*Apolipoprotein E*	*APOE* rs429358, rs7412	ε2, ε3, ε4	Affects repair and plasticity of the brain. APOE isoforms have differing effects on neurite extension, which can influence ability to recover post-concussion.
	rs405509	G/T	Associated with functional regulation of *APOE* transcription.
*Brain-derived neurotrophic factor*	*BDNF* rs6265	Val/Met (C/T)	Affects repair and plasticity of the brain via strengthening existing synaptic connections and modulating the creation of new synapses.
*Catechol-O-methyltransferase*	*COMT* rs4680	Met/ Val (A/G)	Affects cognitive behavioural capacity post-concussion and could increase impulsivity and risk taking.
*Microtubule-associated protein tau*	*MAPT* rs10445337	C/T	Affects repair and plasticity of the brain via modulation of microtubule formation, structural stabilisation of the neuronal axons and drives growth of neurites.
*Endothelial nitric oxide synthase*	*NOS3* rs2070744	C/T	Could affect severity of concussion and cognitive behavioural capacity post-concussion via modulating cerebral blood.

Alleles previously associated with traumatic brain injury are underlined (adapted from Antrobus et al. [14]).

**Table 2 genes-13-00820-t002:** Genotype score of each polymorphism and genotype frequencies in elite rugby athletes and in non-athletes [34].

Gene Name	Gene Abbreviation	Polymorphism	Alleles	Genotype Score	Frequency in Elite Rugby Athletes (%)	Frequency in Non-Athletes (%)
*Ankyrin repeat and kinase domain containing 1*	*ANKK1*	rs1800497	A/G	GG = 2, GA = 1, AA = 0	GG = 65.2, GA = 31.0, AA = 3.8	GG = 65.2, GA = 30.6, AA = 4.2
*Apolipoprotein E*	*APOE*	rs429358 and rs7412 rs405509	ε4+/ε4− G/T	0 = ε4+, 2 = ε4− GG = 2, GT = 1, TT = 0	ε4+ = 28.9, ε4− = 71.1 GG = 25.8, GT = 48.7, TT = 25.5	ε4+ = 28.2, ε4− = 71.8 GG = 26.2, GT = 47.3, TT = 26.5
*Brain-derived neurotrophic factor antisense RNA*	*BDNF-AS*	rs6265	C/T	CC = 2, CT = 1, TT = 0	CC = 67.5, CT = 28.9, TT = 3.6	CC = 66.3, CT = 30.1, TT = 3.6
*Catechol-O-methyltransferase*	*COMT*	rs4680	A/G	AA = 2, GA = 1, GG = 0	AA = 24.8, GA = 49.8, GG = 25.4	AA = 30.2, GA = 47.4, GG =22.4
*Microtubule-associated protein tau*	*MAPT*	rs10445337	C/T	CC = 2, TC = 1, TT = 0	CC = 4.7, TC = 35.7, TT = 59.6	CC = 4.7, TC = 31.4, TT = 63.9
*Endothelial nitric oxide synthase*	*NOS3*	rs2070744	C/T	TT = 2, TC = 1, CC = 0	TT = 37.6, TC = 47.6, CC = 14.8	TT = 38.7, TC = 44.3, CC = 17.0

Alleles previously associated with traumatic brain injury are underlined. ε4+ = ε4 allele possession, ε4− = absence of ε4 allele.

**Table 3 genes-13-00820-t003:** Prior literature-based TGS with kurtosis statistics, and group comparisons via independent *t*-test, top quartile vs. bottom quartile comparisons via χ^2^, and ROC curve analysis AUC.

Group	Mean (SD) TGS	Mean (SE) Kurtosis	*p*-Value Athlete Group vs. Non-Athletes	*p*-Value Top Quartile vs. Bottom Quartile TGS	ROC Curve Analysis AUC (95% CI)	*p*-Value AUC
Non-athletes	56.4 (12.8)	−0.403 (0.217)				
All Rugby Athletes	56.5 (13.6)	−0.506 (0.198)	0.797	0.349	0.504 (0.470–0.538)	0.800
RU Athletes	56.4 (13.4)	−0.490 (0.215)	0.828	0.415	0.504 (0.468–0.539)	0.830
RL Athletes	56.9 (14.7)	−0.617 (0.488)	0.821	0.444	0.507 (0.440–0.575)	0.823
RU Forwards	56.3 (13.3)	−0.384 (0.283)	0.934	0.678	0.502 (0.460–0.544)	0.935
RU Backs	56.5 (13.5)	−0.613 (0.328)	0.769	0.326	0.507 (0.460–0.554)	0.772

## Data Availability

All data underpinning this publication are openly available from the University of Northampton Research Explorer.

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
