# Peer review of "Concussion-Associated Polygenic Profiles of Elite Male Rugby Athletes"

_genes, 2022, doi:10.3390/genes13050820_

Round 1

Reviewer 1 Report

This original research study investigated whether suspected concussion-associated polygenic profiles of elite rugby athletes differed from non-athletes. The authors have made a good effort to address the issues raised and I have no major critical remarks. However, there are some concerns that need to be addressed.

Minor concerns:

  • The TITLE of the article is too long, please make it shorter. The title should be short and informative, such as "Concussion-associated polygenic profiles of elite rugby athletes"
  • INTRODUCTION:
    • Line 97: The authors introduce the term SNP for the first time – it needs expanded abbreviation here.
    • Please provide more information on the selected genes functions and their polymorphisms characteristics or nomenclature. For example, a table could be provided describing the characteristics, functions of genes and their polymorphisms.
    • I would recommend to use the same style for the genomic variant (for SNPs) nomenclature throughout the article.
    • There are some typing mistakes: lines 76, 82, 83, 86, 89, 91, 93, 94, 95. 
  • TABLE 1:
    • Why is one allele underlined in the table 1 (column Alleles)? maybe you need to clarify this below the table.
    • The frequencies of the genotypes are listed in the columns (Frequency in elite rugby athletes (%) and Frequency in nonathletes (%)), but it is not clear what these genotypes are, so it is necessary to indicate which genotypes they were.
  • FIGURE 1 and 2: In this article, the two figures are repeated twice, please remove unnecessary ones.
  • DISCUSSION: Line 379-380:  Genes name should be rewritten in Italics - SPATA5 rs144663795 and PLXNA4 rs117985931

Author Response

Minor concerns:

The TITLE of the article is too long, please make it shorter. The title should be short and informative, such as "Concussion-associated polygenic profiles of elite rugby athletes"

Thank you for this suggestion, we have shortened the title to “Concussion-associated polygenic profiles of elite male rugby athletes"

INTRODUCTION:

Line 97: The authors introduce the term SNP for the first time – it needs expanded abbreviation here.

Thank you for this suggestion, we have expanded the abbreviation.

Please provide more information on the selected genes functions and their polymorphisms characteristics or nomenclature. For example, a table could be provided describing the characteristics, functions of genes and their polymorphisms.

I would recommend to use the same style for the genomic variant (for SNPs) nomenclature throughout the article.

Thank you for this suggestion, we have included a table to provide details of the nomenclature and characteristics of the polymorphisms included in this study. Some polymorphisms are traditionally indicated by either codon or nucleotide, and we have retained that where applicable for the benefit of readers.

There are some typing mistakes: lines 76, 82, 83, 86, 89, 91, 93, 94, 95. 

Thank you for this, the typing mistakes have been corrected.

TABLE 1: Why is one allele underlined in the table 1 (column Alleles)? maybe you need to clarify this below the table.

Sorry if this was not obvious, but we would refer the reviewer to the caption below the original Table 1 (now Table 2) and the statement on original line 172 (now line 214).

The frequencies of the genotypes are listed in the columns (Frequency in elite rugby athletes (%) and Frequency in nonathletes (%)), but it is not clear what these genotypes are, so it is necessary to indicate which genotypes they were.

Thank you for this suggestion, genotypes have been added to the frequencies (%) for athletes and non-athletes in what is now Table 2.

FIGURE 1 and 2: In this article, the two figures are repeated twice, please remove unnecessary ones.

Apologies, they must have somehow appeared as a result of the manuscript submission process. The repeated Figure 1 and 2 should now not appear.

DISCUSSION: Line 379-380:  Genes name should be rewritten in Italics - SPATA5 rs144663795 and PLXNA4 rs117985931

Thank you for spotting that, those gene abbreviations are now in italics. 

Reviewer 2 Report

General comments:

The article „Concussion-associated polygenic profiles of elite male rugby 2 athletes do not differ from non-athletes but the COMT (rs4680) 3 and MAPT (rs10445337) G-C allele combination is advantageous“, provides a novel and practically important data. The article has a multidisciplinary approach, which can be used in general genetic and human movement studies. The introduction is based on relevant literature and provides a good justification of the study’s aim, including the sociological need for such research.

The method section is well described and allows full replication of the study, the sample size is justified and above general standard in sports studies. The statistical analysis is in some part not very clear in the analytical approach between rugby players and rugby union players, this should be presented more clearly e.g. also as part of the figure legend.

Discussion is the strongest part of the article, thus I have no more points to suggest.  The authors might mention some distinguishing factors in elite athletes such as strength-related genes, PPAR´s. There might be a discussion about whether higher frequency or severity of a concussion is more dangerous for athletes and might have genetic relation.

Specific comments:

Line 38: The analysis of players’ position is not mentioned in the study aim. Amend it.

Line 49: Some keywords are already in the title, use more general ones.

Line 43: Performance level should be mentioned in the aim as well.

Line 61: The possible genetic relation should be mentioned already in the introduction.

Line 79: at this point or in the limitation section, the problem of concussion severity should be mentioned.

Line 194: Which analyses showed the differences in rs405509, rs4680 and NOS3 rs2070744?

Line 291: The rugby athletes and RU athletes analyses should be mentioned in the statistical section.

Author Response

Thank you for the review of our article. In response to the latter comment, we now mention the issue of whether incidence or severity of concussions is more problematic within the limitations section of the Discussion.

Specific comments:

Line 38: The analysis of players’ position is not mentioned in the study aim. Amend it.

Thank you for this suggestion, we have amended the aim to include analysis of playing position “between rugby union forwards and backs”.

Line 49: Some keywords are already in the title, use more general ones.

Thank you for the comment. As requested by the other reviewer, we have shortened the title to “Concussion-associated polygenic profiles of elite male rugby athletes" so have not changed the keywords because there is less repetition.

Line 43: Performance level should be mentioned in the aim as well.

Thank you, you are right that lines 43-44 refer to elite status. However, this study only included elite athletes and non-athletes, and did not include lower level competitors, so we do not think performance level would be appropriate in the aim. We hope you accept our rationale.

Line 61: The possible genetic relation should be mentioned already in the introduction.

Thank you for this comment, we now refer to the possible genetic relation at the end of the first paragraph of the introduction, before expanding on that later in that section.

Line 79: at this point or in the limitation section, the problem of concussion severity should be mentioned.

Thank you for this comment. We now refer to the issues of concussion severity and incidence in the final paragraph of the discussion.

Line 194: Which analyses showed the differences in rs405509, rs4680 and NOS3 rs2070744?

Sorry, we can see that this was not clear. We have adjusted the text to read “In addition, a TGS algorithm determined only by observed genotype frequencies in elite rugby athletes was also calculated, wherein three of the seven genotype scores allocated had different values that reflected the prior evidence [34]” to better explain that this was a part of the method of developing two different TGS algorithms, and not a statistical result.

Line 291: The rugby athletes and RU athletes analyses should be mentioned in the statistical section.

Apologises for the lack of clarity. We have extended the penultimate sentence of section 2.4 to name the groups that were compared using multifactor dimensionality reduction, which are subsequently shown in Figure 3.
